# ICT in Rural Areas from the Perspective of Dairy Farming: A Systematic Review

Alba Vázquez-López *, Martín Barrasa-Rioja and Manuel Marey-Perez

Grupo de Investigación PROePLA, Escola Politécnica Superior, Universidad de Santiago de Compostela, Campus Universitario s/n, 27002 Lugo, Spain; martin.barrasa@usc.es (M.B.-R.); manuel.marey@usc.es (M.M.-P.)
* Correspondence: albavazquez.lopez@usc.es

**Abstract:** This study presents a systematic review of 169 papers concerning the ICT (Information and Communication Technologies) related to rural areas, specifically to dairy farms. The objective was to delve into the relationship between dairy farmers and the administrative authorities via e-government, comparing this area to another eight concerning the farmer's needs and expectations in relation to the ICT in different fields of their business. We observed that areas such as connectivity and digital inclusion are the most covered areas not only at the study level but also at the government level since countries all over the world are trying to develop politics to put an end to the so-called "digital divide," which affects rural areas more intensely. This is increasing due to the growing technological innovations. The areas of the market, production, financial development, management and counseling, Smart Farming, and Internet of Things have been approached, associated with the ICT in dairy farms, showing in the latter two an increasing number of papers in the last few years. The area of public administration in relation to dairy farms has also been covered, being remarkable the low number of pieces of research concerning the interaction by the farmers, more specifically by dairy farmers, with the public administration, which is surprising due to the new global need and especially in the European Union (EU) of interacting with it telematically by all legal entities. The results show that there are still barriers to the implementation of the electronic government (e-government) since the websites do not meet the user's expectations. Therefore, this study lays the ground for future research on this area. As a graphical abstract of the contributions of this paper, we present a graphic summary, where the different contributions by areas and expressed in percentage values are shown.

**Keywords:** dairy farms; ICT; e-government; smart farming; digital divide

## 1. Introduction

The Internet has become an essential tool for communication, education, entertainment, shopping, and many other applications. In fact, nowadays the access to cyberspace is not a luxury any longer but a necessity [1]. This has fostered what is known as the "Fourth Industrial Revolution" or "Industry 4.0," propelled by the appearance and progressive maturity of the ICT applied to industrial processes and products [2]. This revolution has brought important progress and social and economic benefits but also has led to a new type of inequality often known as the "digital divide" [3,4], tries to put an end to this situation through the promotion of the rapid and ultra-rapid online access for everybody, as well as carrying out strategies which lead to digital literacy, skills, and inclusion [5]. Nowadays, this tendency to digitize and automate processes with tools such as Smart Manufacturing, Smart Production, or Industrial Internet [6] makes it essential that everybody knows how to use the ICT effectively, both for their economic benefit and for their life quality [7].

In the case of the production of foodstuff, the current concern related to global alimentary security, climate change, animal welfare, biodiversity, and natural resources disposal highlights the necessity of sustainable development of agriculture, which is only possible

with the support of the ICT [8]. Within this field, it is expected that the new technologies fill the gaps, complement and improve the rural services and reduce the rural inhabitants' isolation, creating cohesion and opportunities [9,10], thus that they are able to work remotely and interact both with their local and global communities in different ways [11]. To authors such as [12,13], the future of livestock goes together with sustainability. It is precisely here where digitalization becomes a key aspect since apart from meaning an improvement in the production of farms, it can facilitate the exchange of experiences between the farmers themselves, experts, and other interested parties. In the last years, dairy farms have been subjected to an increment of the demands in terms of quality, together with a reduction in the price of their product, which meant the necessity to increase the efficiency of the production. It is at this point where the adoption of the "Precision Agriculture" comes into play, with the use of "Internet of Things" or "Big Data Analytics" [14–18]. It is also important to take into account the current state of the art of the dairy industry, where two main standpoints can be distinguished: The Supply Food Chain (SFSC), where there is a very direct approach to the client [19–22] or the great organized distributions [23–25]. Regarding the use of ICT in these terms, it is worth mentioning studies conducted by [20] in Baja California (Mexico) of manufacturing companies based on SFSC, where they observed eight main factors for making decisions with the use of new technologies: Facilitation of information, operation management, production control, visibility among partners within the supply chain, operation management in the manufacturing lines, flexibilization, low costs, and short cycles with clients. In addition, [21] along this line, in their studies in Spain of agroecological cooperatives show the importance of the ICT in the relational web of the different agents involved in the supply chain. In large supply chains, according to [23], information and communication technologies and the Internet of Things allow to efficiently carry out the traceability through the whole supply chain of agricultural products eliminating the problems of security, quality, and defense of agricultural products, since they give precise and real-time information, information which is transparent and reliable from the field to the plate.

The present paper focuses precisely on the last point: Dairy livestock in the field of the ICT. It consists in performing a systematic review of the existing literature about the relation between the dairy farmers and the ICT within the different fields of their management, with the aim of knowing the current state of the art in this area and identifying the existing knowledge gap to lay the ground for future research.

## 2. Methodology

First, a systematic search using keywords and concepts related to academic disciplines, which addresses the topic of connectivity and inclusion of the ICT in rural areas and dairy livestock within the different fields connected with their activity, was performed. The chosen terms related to inclusion and connectivity were: Digital development, digital divide, ICT, Industry 4.0, internet adoption, internet use, technological isolation, web accessibility. Related to the use of the internet in dairy farms: Administration procedures, animal identification, animal procedures, legislation, business procedures, dairy farm *, dairy farm business, dairy farmer *, e-government, farm procedure, farm * development, farm * technology, government procedures, ICT advisory, legal regulations, new concept, public administration, rural areas, rural communication, rural development, rural employment, rural internet, rural market, rural network, smart farm, and state regulation.

The search, with combinations of these 8 ICT terms and 26 rural terms, generated a total of 56 search queries in the databases of Scopus, Google Scholar, and Web of Science. The queries for the aforementioned database were reduced due to their wide scope and the variety of academic disciplines, and they were refined, only choosing those academic disciplines which were relevant for the aim of this work, that is to say, those which were in charge of the digitalization and its implementation in the rural world, more concretely in dairy farms. Through this selection, we focused mainly on papers of developed countries, chiefly European. The search queries identified a total of 1683 potential papers. Eventually,

169 papers fulfilled the final restrictions adopted, to which we have to add their scientific level and newness.

## 3. Results

In order to provide a clear and ordered representation of the results, we generated a diagram of dairy farms in relation to the farmers' expectations and needs concerning the ICT in all the areas connected to their business (Figure 1). This selection led to the division of this paper into nine thematic areas. The main focus was the dairy farmers' relation with the Administration.

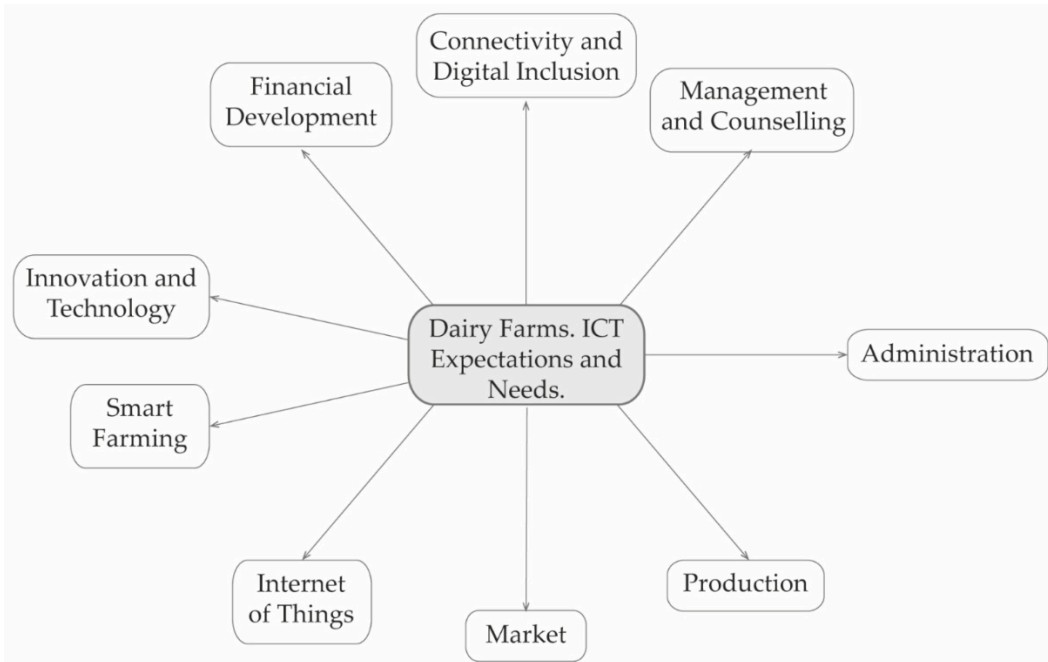

**Figure 1.** Conceptual map of dairy farmer's expectations and needs.

### 3.1. Connectivity and Digital Inclusion (35 Documents)

Connectivity was defined within the objectives of the European Union as the access to the high-speed web from any place [26]. The term "digital inclusion" was defined as each individual's degree of use of the different options and opportunities offered by the internet [27,28]. In our current society, digital inclusion has become a key factor in terms of social equality because of the enormous advantages that the internet offers [29].

In spite of the importance of the ICT for social inclusion, economy, interpersonal bonds, education, the army, politics, and culture [1,30–33], there are still problems associated with the lack of connectivity and inclusion, which imply a new type of inequality called the "digital divide" [3,34–37]. These inequalities are internationally revealed among countries due to their different economic development, and within countries, through different social groups [38–40], affecting mainly those rural areas which have problems with the implementation of ultra-rapid internet [41,42].

According to studies performed by [5,34,43–49], two main lines of research can be distinguished in this field: Research about connectivity, which shows that there exist growing and persistent differences in data infrastructure quality between urban and rural areas, and research of inclusion, showing that the difficult diffusion of technologies and the average lower levels of education and skills in rural areas have a negative impact on the adoption and use. This is evidenced in research carried out by [35] in the province of Groningen, in the north of the Netherlands, as they show examples of the necessity of high-quality bandwidth in rural areas. However, they also show that many rural citizens still present problems related to the lack of skills, capacities, and perseverance required by

the ICT, which lead to the generation of disinterest, and as a consequence, to the lack of initiative in terms of bandwidth in some Dutch "blank areas."

Another clear example of these differences between the rural and the urban world can be seen in field studies developed by [50] in rural communities in Australia, namely in Boorowa and Crookwell (in New South Wales), where it is shown that in spite of the fact that the rural inhabitants understand and are willing to accept certain disadvantages (such as having to commute to urban nucleuses in order to do the shopping), the low-quality or inexistent internet connection and at higher prices than in urban areas is not considered just a disadvantage, but a problem as serious as the lack of water or electricity. In fact, they show several examples of how these connection issues affect them in every aspect of their daily life: From education to health or businesses, among others. To authors such as [50–52], the rural digital exclusion is an unsolved problem which, in fact, is still increasing, since even though great amounts of resources have been intended to improve the infrastructure and foster digital participation, there still exists a number of problems related to the lowest levels of digital skills and knowledge, as well as fewer opportunities to develop these skills due to a bad data infrastructure, as made clear by [48,53].

There are several coexisting factors related to the lack of skills: Demographic issues, income, age, level of education and digital literacy [30,54–57]. As an illustration of this situation, semi-structured interviews led by [30] in rural areas of Scotland show that despite the importance that the ICT has for rural businesses, there exist serious problems associated with low-quality or inexistent bandwidth infrastructures, as well as the rural inhabitants' lack of digital skills in several cases. This situation of exclusion, as demonstrated in studies performed by [44], leads us to think about the importance of exploring why people, in spite of having a good connection, do not log in.

*3.2. Financial Development (15 Documents)*

The application of the new technologies within the business field is marking the beginning of a new era known as the "Fourth Industrial Revolution" or "Industry 4.0" [58]. This fact is arousing a huge interest in the companies since it promises to increase the productivity, flexibility, and automation of the internal processes, integrating value chains and supporting the companies in designing and offering new services based on the availability of data enabled by different technologies [59–61]. Enterprises are investing big efforts to understand how these technologies could be implemented to take advantage of their actual operations and offer a more competitive value proposition to the new and current clients [62]. As an appreciation of this tendency to the incorporation of the new technologies both at a social and at a business level, studies performed by [63] in 72 countries worldwide, affirm that the ICT are essential nowadays both for the individual and for companies, becoming a key tool for social inclusion and financial development, and contributing to the reduction of the poverty of any country. Another example of that can be seen in a study carried out by [64], where from the analysis of data collected from Eurostat of a survey about the use of the ICT at a household level in 2017, many of the benefits with which people are provided nowadays were reflected, such as the buying and selling of goods and services and the reduction of the time needed to complete any administrative procedure.

This tendency also offers many advantages to micro-companies, particularly to those based in rural areas, despite the fact that digital divisions, both social and economic or even territorial, continue to create challenges for this sector of the rural economy [56]. In this field, more concretely in that of the production of milk, as is shown in research conducted by [12] in Baakse Beek (east of the Netherlands), the digital economy, within the frame of Industry 4.0, has an enormous application, since it has had great success in the increase of the productivity of the farms, achieving a sustainable economic expansion. Thus, other studies such as those performed by [13], in which drawing from surveys from the Irish Farm Accountancy Data Network (FADN) in Ireland, it is shown how the economic profits of the farms depend on their own level of innovation. Therefore, innovative farmers can obtain higher economic profits, recovering the whole investment made in innovation.

However, as demonstrated by [65] in their research on dairy farms in the United Kingdom, the ICT also imply meaningful changes in the knowledge and skills required from the farmers, as in the case of Automatic Milking Systems (AMS), which imply that farmers need to learn how to control and use as much electronic data as possible. In this context, it is essential to remember that until recently, these users had a limited control over their exposition to the risk factors—the price of the milk or the seasonal conditions [66], which have been progressively reduced by the application of the ICT.

Another key factor in terms of livestock development is the change of paradigm in terms of the support to farmers through the PAC (Common Agricultural Politicy), which is being constantly reduced and makes it necessary for farmers to have a high degree of knowledge about the SWOT (Strengths, Weaknesses, Opportunities, and Threats) of their farms in order to achieve the objectives of the market. As demonstrated by [67] in their studies performed in Central and East Europe (Lithuania, Poland and Slovenia), clear differences both at a strategic level and at an agricultural objectives level among the analyzed countries can be observed. As a result, the farmers' attitudes towards the market and the future Common Agricultural Policy are even more negative than those of a group of farmers from West Europe.

In terms of the current general situation of farmers throughout the world, it is important to highlight that they are facing an increasingly turbulent business environment as a result of the rise of the volatility of the price of the milk, as reflected in studies conducted by [68] from databases of dairy producers in New Zealand. As a consequence of these global conditions, the use of the ICT in the farms has become fundamental, for it works as a method of reducing the risks and the uncertainties, and it allows the making of more enriched and informed prescriptive decisions regarding energetic efficiency, productivity, commercialization, and diagnosis, which are essential to remaining in the market [2].

*3.3. Innovation and Technology (22 Documents)*

An innovation is an idea, practice, or object which is perceived by an individual as new [69]. Nowadays, technological innovation is a central engine for the economic growth and productivity of any country [70], since it has helped satisfy the growing demand of consumers, which itself is greatly propelled by the innovations in consumption technologies and commercial practices [71]. The capacity to innovate is a strategic tool for those companies which want to keep their competitive position in the global market [28,70,72]. It is precisely here where the concept of Industry 4.0 emerges, which is a movement of global modernization in the manufacturing industry towards the adoption of the recent progress in the field of the ICT: New communication systems and protocols, online security standards, multiple-devices displays, mobile and compact communication devices with increasingly better computational capacities [6,16]. In terms of this innovation capacity offered by the ICT, as shown by [73] in their research performed in Beijing or in other studies such as those conducted by [74] in Igloolik (Nunavut, in the north of Canada), it is demonstrated that the new technologies are increasing the human capacity of information storage, analysis and communication, which means that more people have the opportunity to produce, communicate and use digital media as never before.

The agro-alimentary sector is the biggest manufacturing sector within the EU, and it is one of the main economic boosters of the member countries [75]. In this field, the incorporation of the new information and communication technologies to the productive chain results in the improvement of the competitiveness of the sector [76,77], apart from allowing people to live and work remotely and to be able to interact with their local communities in different ways [11]. At this point, the concept of "Smart Agriculture" comes into play, and it is constituted by farms and farmers, which use the progress in the information and communication technologies in order to react to challenges [57]. As an example of that, a case study performed by [78] in Ireland shows that there exists a positive relationship between the development of the competition within the Irish agricultural sector (which is achieved through the participation in the formal agricultural education)

and the use of innovations, technology, and better management practices since they involve an improvement in the level of income of the farm.

However, and as has been said, in spite of the fact that digitalization, automation, and artificial intelligence are increasingly important to farmers, researchers, and people responsible for the decision-making process in the field of food and agriculture [10,79], there are still problems related to their use and implementation [80]. Thus, and connected to this, interviews conducted by [81] to educated young farmers in the west of Ireland show that in order to achieve a correct implementation of the new technologies, it is necessary to adapt the messages and methods to fulfill the interests and objectives of the users' innovations. At this point, it is also important to mention the farmers' point of view towards the opportunities perceived as feasible to obtain commercial income, which makes it fundamental to be conscious of the role of the farmer as an entrepreneur actor in the strategic decision-making process, as reflected by studies carried out by [82] about Dutch dairy producers. Particularly, other research such as conducted by [83] (using data from the Irish National Farm Survey) reveals that, in spite of the fact that the use of the internet is positively associated with the income and performance of the farm, farmers' use rates are still quite low. For an improvement of implementation in this concrete field, studies published by [84] about farmers in New Zealand assure that it is necessary to incorporate flexible and simple tools that take into account the user's needs and allow the implementation of other tools.

*3.4. Smart Farming (12 Documents)*

The term "Smart Farming", as defined by [85,86], tries to incorporate advanced technologies such as sensors, artificial intelligence, and robotics to farms as a medium of increasing the efficiency of the production of food and to minimize the use of resources. The tasks of farm management, with the use of these technologies, are not only based on the positioning, but also take into account improved data and the context in real-time [15]. From the farmers' point of view, Smart Farming should provide them with an assured value in the form of a better making of decisions for a more efficient farm business management [87], since due to the market demands, it is necessary to optimize the management to achieve a better quality of the product [88]. In this way, the application of these new technologies to farms allows the development of specific models for them, providing the individual farmer with the capacity to plan their activities in response to the changing circumstances [15,89,90].

In this field, it is important to highlight that research led by [91] in Italy, reflected the acknowledgement from the farmers towards the application of the new technologies as the only way to remain competitive and to improve sustainability. It is worth mentioning that interviews conducted by [92] to 817 German farmers showed clear evidence of the importance of technology (more concretely of smart phones) in agriculture as facilitating elements which also improved many operative procedures, as it can be combined with agricultural accuracy technologies. Despite all the aforementioned evidence, the study also shows that the rate of the use of technology is still lower than expected since there are factors that hinder the process of implementation, such as age, education, or the size of the farm. Therefore, due to this clear tendency to the incorporation of technologies in this field, interventions at niche level will be necessary to improve the farmers' organism and their local networks in these transactions, as well as the cooperative design of new institutions at a regime level in order to facilitate the fair and transparent assignation of risk and benefit in the information of agricultural data, thus being able to fulfil the demands of the market, as shown in studies performed by [18] of 28 farmers in Australia.

Hand in hand with "Smart Farming", the phenomenon of "Big Data" appears: Massive volumes of data covering a wide range of fields, which can be captured, analyzed, and used in the decision-making process [17]. These are being used to provide predictive information in agricultural operations, to propel operative decisions in real time and to redesign commercial processes, playing a key role in the current networks of the food

supply chain [85,93]. In fact, more precisely in the area of dairy production, they are contributing to the improvement of productivity and profitability [18].

### 3.5. Internet of Things (16 Documents)

With the arrival of the Internet of Things (IoT), the scope to the development in the field of agriculture and the supply chain has increased considerably [94,95]. IoT is the next step of the internet, in which even physical things are connected to one another [96,97]. This concept combines "Internet" and "Things," therefore, it can be semantically defined as "a global network of interconnected objects with a unique direction, based on protocols of standard communications" [98].

In this way, IoT provides new levels of visibility, agility, and adaptability of the supply chain to face various challenges of SCM (Supply Chain Management). The data emitted by the smart objects, once compiled, analyzed, and turned into useful information in an effective manner, can offer unprecedented visibility in all aspects of the supply chain, yielding early alerts of internal and external situations, which require a remedy, and implies a reduction of the costs, and, therefore, a significant improvement of the productive efficiency [14,99,100]. This has an enormous potential of application in the field of food and agriculture, especially in view of the social and climatic challenges that this sector face, since they could transform it, contributing to food safety and to the reduction of agricultural consumables and food waste [101]. In the case of farms, the use of GPS for the control of the herd is becoming essential [102–104], in the same way as in the case of virtual fences as a method of controlling animals—without visible, physical fences—and tracking their state [105–108].

### 3.6. Market (12 Documents)

Due to the great importance of the ICT to the economic and social development of any country, the European Commission declared a strategy of Digital Single Market (DSM) as one of the 10 main priorities, committing 61.300 million euros to the budget period from 2014 to 2020 [109]. It is here where we can see the importance of the new technologies for rural businesses and activities that have to face the problems of spread population, the distance between clients and suppliers and communication costs and delays. In this way, the electronic network and commerce have become some of the most important factors for small rural companies [110,111]. Going deeper into the field of the application of the new technologies in the rural world, it is essential to highlight that they serve to promote innovation in the farms, improve the rural context, allow a more sustainable rural development, and foster an economic growth [112,113]. Along this line, the use of the ICT itself or combined with other ICT systems in the farms has resulted in an improvement of the productivity and better use of the resources, reducing the necessary time in terms of marketing, logistics, and quality assurance, which implies a better making of decisions and, therefore, a great improvement of the conditions of market effectiveness [87,114].

Specifically dealing with the dairy field, it is worth mentioning that the current tendencies of the market, added to the abolition of dairy fees, have accelerated the tendency of the European dairy industry towards bigger farms and more productive cattle [115,116]. At this point, it must be highlighted that the ICT are becoming crucial in the application of prognostics, such as the price of the milk or pastures since this sector is characterized by a volatile pattern of demand, influenced by an environment of a quick and dynamic response [117]. In the field of application of the ICT to this sector, research developed by [118] in Flanders (Belgium) and Northern Ireland (United Kingdom) demonstrates the importance of the creation and exchange of knowledge with the support of innovative technologies in order to face complex challenges such as the climate change, depletion of resources, global markets, and the changing social and legislative prospects, thus that they become an essential condition to stimulate sustainable agricultural production. In terms of sustainable agriculture, surveys elaborated by [116] in Greece show the great importance of this terminology in a sector in crisis, as is the case of the dairy industry. In fact, they remark

the importance of political interventions oriented towards the creation, management, and governance of the producers' organizations, since a strong and feasible organization will reduce the production and transaction costs, increase the farmer's negotiation power, and contribute to a balanced distribution of power among the chain actors.

### 3.7. Production (18 Documents)

The growing urban-rural inequalities and the accelerated depopulation have become a general concern in the fight against the rural decline and urbanization all over the world [119], more concretely, the new course of the politics in the EU is increasingly oriented towards the promotion of small agro-zootechnical productions from a sustainable point of view, fostering the return to agricultural activities by young people [120].

In terms of the agricultural development, as demonstrated in surveys carried out by [121] to Irish farmers, it has always affected—and has been affected by—the context in which it operates, both in the past and in modern days; in such a way that three clear business dimensions by the farmers can be distinguished: Diversification (for example, care, education, and farmers' market), finalization (that is to say, abandonment of the dairy sector), and maximization (the use of consumables which are external to the farm in order to maximize the efficiency per hectare) of the production. Concretely, in terms of milk production, it has been seen that it has drastically increased during the second half of the XX century, as a result of the agricultural modernization, which stimulated the specialization, intensification, and scale enlargement [73,122,123], which makes the extension of sustainable agriculture necessary [124]. Currently, in this field, there is a growing demand at a global level of dairy products, which brings deep changes in this industry [122,125,126], giving room to the opening of new market niches, where increasing the efficiency of milk production is key to survive in the market [14]. All this has brought a growing need of knowledge about the usefulness of the information and communication technologies in order to achieve the goals proposed by the market of sustainable agriculture, that is to say, to improve the agricultural productiveness and increase the incomes [79,99,127–129]. To this effect, strategies focused on the creation of the value chain, transparency, and traceability will also be necessary [116].

Dealing with the number and type of farm businesses, according to studies conducted by [130] from the analysis of data of dairy farms provided by Farm Business Survey of England and Wales (FBS), it is observed that in spite of the high efficiency of milk per cow in many industrialized countries in the European Union, as is the case in Wales and England, the dairy production continues with a long-term tendency to the reduction in the number of farm businesses and intensification, which is fostered by socio-economic and political aspects. In this context of the reduction and intensification of dairy farms, other surveys such as those elaborated by [131] in Australia show the importance of guaranteeing the best practices with the use of the ICT to all those farmers who consider expanding their business in terms of cattle management, operations, finances, human resources, and strategic administration.

### 3.8. Management and Counselling (8 Documents)

In terms of the complexity level of management and counseling for agricultural companies, it has been gradually increasing during the last few decades, from simple units of production, which provided the population with enough and affordable amounts of food; to agricultural businesses with multifunctional service sectors [132]. Here, two clearly differentiated sides can be distinguished: Those who support the idea of cows going outside the farms and those who claim that cows should only be fed within the farms [133]. In this way, and due to the increasing complexity of the farms, in the organizational system, the adhesion to the cooperatives is becoming more common in order to be able to satisfy the market needs since they are providing their partners with joint operations and fostering environmental sustainability, representing one of the most common organizational models of provision of services for small farms, as can be seen in personal interviews carried out

by [134] in three Galician farms associated with cooperatives with an obvious positive relation between the association of cooperatives and the reduction in the carbon footprint of the farms. Related to all that, studies performed by [135] about dairy producers and members of cooperatives show this obvious relevant role of the cooperatives themselves in the generation of confidence to milk producers to be able to fulfill a growing market demand. Besides, as reflected in this study, the only way to adhere to this market is through collaborative communication between cooperatives and users (exchanging information, discussing expectations, and providing credible information about the relationship between the offered price and the milk quality).

Regarding the success of the farms, there is clear evidence that it will increasingly depend on the manager's capacity to develop a skilled, motivated, and passionate working force, as demonstrated in telephonic interviews elaborated by [136] to farm workers from the United States of America. In this way, it can be affirmed that within the current competitive environment, a farm can survive in economic terms and be sustainable when it is well managed [117]. This can be seen in research conducted by [78] in Irish farms, where participation in formal agricultural education is essential, fostering the use of innovations, technologies, and better management practices, for it brings a positive impact in terms of farm income.

More specifically, in the area of personalized counseling to farms, it is crucial to take into account the importance of experience and longevity of companies to build trust to farmers [137]. Currently, there exist agricultural Financial Management Information Systems (FMIS) to support the management of several agricultural companies; it consists essentially in a Management Information System (MIS) for the agricultural sector as an organizational method in order to provide past, present, and projected information related to internal operations and external intelligence [138].

*3.9. Administration (36 Documents)*

Nowadays, the [139], acknowledges and underlines the dependence on digital media in both the public and private professional and personal daily life [140]. This situation is translated as the increase of the offered services by the new technologies, such as online webs [34,37,141], where the users access daily a great variety of internet functions, for example, to search for information, communicate with the government, read news, and buy and sell products [142]. All this shows obvious evidence of the fundamental role which the internet plays concerning natural resources and education [53] since it has reorganized the actions and economic, social, and political relationships worldwide, provoking an "information revolution" [143,144].

This situation is also transferred to the public sphere, for the basis of the new digital economy is remodeling the government through new forms of communication and public social services [145,146]. As an example of that, research developed by [29] in the United Kingdom highlights a growing dependence on the internet, both at a public and private level, feeding the argument that broadband ought to be universally available for everyone, and those who reside in rural areas should not be discriminated in terms of digital connection. In this same field, other studies such as those performed by [50] in Boorowa and Crookwell in New South Wales, Australia, demonstrate that digital involvement is becoming increasingly essential as governments adopt the "digital first" as their main provision of services in spite of the fact that this situation is negatively affecting the rural users of the internet, who experiment online spaces differently as a result of the deprivation of services. Therefore, as shown by studies performed by [147] in Spain, the websites of the public administration are key tools for the diffusion of data coming from the public sector, for they have a direct impact on the satisfaction and trust of the general public. In this field, it is important to highlight that the policy of the European Union towards the convergence of electronic government systems of the member states has as an objective the creation of an integrated and well-established place of work both for the citizens and for the companies or juridical entities at any rate [148].

In spite of that, there still exist critical factors of success, barriers to the adoption of the electronic government and of technology in general, which make clear the fact that the current models do not fulfill the citizen's needs and expectations [149,150]. Related to this situation, we find the scarce involvement of the possible users in the designing process [151]. At this point, the theories of technology acceptance come into play, such as TAM (Technology Acceptance Model), which is based on three fundamental factors when it comes to use decision: Perceived usefulness, the ease of use, and the perceived enjoyment [152]. On the basis of this theory, several studies were developed, such as those led by [153,154]. In research carried out by [153] in China, it has been observed that specifically related to the use of the electronic government, there are differences between urban and rural areas, being fundamental in the latter in terms of acceptance: Ease of use, perceived usefulness, perceived security, the subjective norm, and the inhibiting factor. In their studies, ref. [154] connect confidence to the level of participation, being fundamental to it: Transparency, responsibility, and response capacity. After TAM, there have been other models such as UTAUT (Unified Theory of Acceptance and Use of Technology) [155] in their research in China, refer to the importance of confidence to the acceptance of technology, specifically the importance of politic confidence. Ref. [156] take as the main factors: Performance expectation, effort expectation, social influence, and facilitating conditions. Ref. [157] in their studies concerning the use of a European electronic platform, show as success factors: The habits and facilitating conditions. After this model, a new one called UMEGA (Unified Model of Electronic Government Adoption) appears, which takes as determinant factors to the electronic participation: Effort expectation, social influence, facilitating conditions, perceived risk, use attitude, and behavioral intention to use [158]. Ref. [159] Based on the theories UTAUT and UMEGA, highlight four crucial factors to the adoption of technology: Provision of conditions to facilitate the e-government, the scope of the risk, the nature of the perceived quality of the service, and the trust in the government.

However, as mentioned before, e-government services are increasing, as shown by investigations led by [37] in rural areas in Helsinki, Finland. There are clear cultural and socioeconomic differences, which can be extrapolated among countries and regions, provoking several patterns of adoption and use of electronic services at national and local level. Considering that, and according to studies done by [160] regarding the use of electronic services by the citizens of peripheral regions, there is evidence that, currently, these are less likely to use the several functionalities of these electronic headquarters, even though they are the most benefited from them because of the geographical challenges they face. It is worth highlighting that currently all governments work towards initiatives, which allow the use of web platforms via smartphones originating the so-called m-government [161–166]. Ref. [163] In their studies in rural areas in Tanzania detected five main factors of the intention of use of the m-government: The competitive advantage, the ease of use, the compatibility, the support for the government, and the demography. Authors such as [164] in their research based on UTAUT and UMEGA, carried out in Harbin (China), found gender, age, and education as main factors. Thus as to achieve a correct development of web applications by the government, it is necessary to take into account that these services are used by a wide variety of users' profiles (such as students, workers, youngsters or seniors), who also present different needs, which makes it essential to adopt a segmented approach to study and understand these needs in order to maximize the adoption potential [160,167].

In terms of the concrete field of the relationship existing between farms and the pertinent administrative authorities, it is key to highlight that this relationship is often affected by the inefficient use of the ICT, since lowest-income populations are disadvantaged in terms of connectivity and, therefore, the inclusion of the new technologies in their activity [38]. Closely related to all this, surveys performed by [168] to German dairy producers reveal that these assess the electronic functions associated with animal health care, the reproduction management, and data compilation as the most useful ones. Moreover, they highlight the user-friendliness and the perceived usefulness as key components for the

acceptance of technology, which at the same time have a positive influence on the intention of using such applications. Within this same field, other studies such as those conducted by [169] to dairy producers in Austria show that the farmers' current concern towards the implementation of the ICT in their activity is to have all the relevant data about the animals available through a central database, which makes obvious the necessity to develop an integrated tool of data management.

## 4. Discussion

The systematic review of the literature provides a general vision of the importance of the ICT in the current society, more concretely in the rural world in the field of dairy livestock. In spite of the fact that this sector is fragmented into several disciplines and matters, we have extracted the predominant ideas about how dairy livestock is being affected by the socioeconomic changes of the last few years. Together, the 169 papers reveal that society as a whole is moving towards a digital information society.

The topics that have been more intensely worked upon are those referring to innovation and technology (Table 1), connectivity, and the inclusion of all people. In order to do it, the idea that all farmers should know how to effectively use the ICT in order to obtain economic benefits and improve their life quality is becoming stronger. The concepts of Smart Farming and IoT have also become increasingly important, and this can be observed in the appearance of numerous papers referring to them and also in the intensity with which they are being worked upon. With the appearance of the two aforementioned concepts, the use of the ICT is becoming essential as a method of reducing the risks and uncertainties in the field of dairy livestock [2]. Another sector in which the ICT is becoming a key tool is within the management counseling of farms, as the success of the new typologies of dairy farms greatly depends in the manager's capacity to develop a skilled, motivated, and passionate working force, which is only possible through the implementation and use of the ICT [136].

**Table 1.** List of articles used by thematic areas.

| Thematic | Number of Papers Found | References |
|---|---|---|
| Connectivity and Inclusion | 35 | [1,3,5,26–57] |
| Financial Development | 15 | [2,12,13,56,58–68] |
| Innovation and Technology | 22 | [6,10,11,16,28,69–84] |
| Smart Farming | 12 | [15,16,18,85–93] |
| IoT | 16 | [14,94–108] |
| Market | 12 | [87,109–118] |
| Production | 18 | [14,73,79,99,106,116,119–131] |
| Management and Councelling | 8 | [78,117,133–138] |
| Administration | 36 | [29,34,37,38,50,53,140–169] |

In terms of the field of public administration, it is important to take into account that in order for the rural inhabitants to use the web pages, elements such as the ease of use or the perceived usefulness are fundamental [153]. Moreover, trust [154], especially in the government and in the nature of the perceived quality of the service, are key tools as well [155,159]. The current reality puts us in an unresolved problem of e-government adoption, as the web pages still do not meet the citizens' expectations [150], being the rural inhabitants the least prone to use them [160]. As a result, we believe that the elaboration of studies focused on rural areas has an exponential value and relevance nowadays, since it is essential to take into account the importance of the rural world, more concretely of dairy farms, in the production and supply of the worldwide population. These sectors are obliged, by a new administrative procedure law at the European level, to communicate with the administration telematically. In order to do this, knowledge and education in the field of the ICT have become a necessity. At this point, it is worth asking: What is this relationship farmer-administration like? Do the current electronic headquarters and

applications fulfill the users' necessities and expectations? How close and friendly are they for them? We will try to answer these questions in future research.

## 5. Conclusions

This systematic review of the literature has served to delimit and clarify the dairy farmers' expectations and needs concerning the ICT in the different areas of their business. As we could observe, there is a knowledge gap in terms of the relationship between dairy farmers and the administrative authorities through the use of ICT. These findings open new research lines oriented to know: Why do these users—despite having an internet connection—not use e-government? What factors influence the acceptance of these web platforms? What are the problems detected by those who do use them? In future research, we will aim to answer these questions.

**Author Contributions:** Conceptualization, A.V.-L., M.M.-P. and M.B.-R.; methodology, A.V.-L., M.M.-P. and M.B.-R.; validation A.V.-L., M.M.-P. and M.B.-R.; formal analysis A.V.-L., M.M.-P. and M.B.-R.; research A.V.-L., M.M.-P. and M.B.-R.; data curation, A.V.-L., M.M.-P. and M.B.-R.; writing—original draft preparation A.V.-L., M.M.-P. and M.B.-R.; writing—review and editing A.V.-L., M.M.-P. and M.B.-R. All authors have read and agreed to the published version of the manuscript.

**Funding:** We would like to thank the Galician Government (Xunta de Galicia) for their financial support (IN607B: Modalidad B-Grupos con potencial crecemento (GPC GI-1716 PROXECTOS EPLANIFICACIÓN (PROEPLA) (2018-PG078) Ref.ED431B 2018/23)).

**Data Availability Statement:** All data were extracted from public sources, mostly from scientific papers.

**Acknowledgments:** The authors would like to express their gratitude to Xoel Rodríguez Rodríguez for his labor translating this paper.

**Conflicts of Interest:** The authors declare no conflict of interest.

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
