# Peer review of "ICT in Rural Areas from the Perspective of Dairy Farming: A Systematic Review"

_futureinternet, doi:10.3390/fi13040099_

Round 1

Reviewer 1 Report

ICT in rural areas.

Dairy farmers´ procedures with the e-government: a systematic review.

The manuscript provides a literature review of ICT in rural areas with focus on dairy farming.

Strengths: The manuscript review many relevant papers.

Weaknesses:  Methodology needs to be clarified. See below.  Issues with Table 1 (see below).  Table 1 impacts Figure 1 (see below). Manuscript needs proofreading for proper use of English language.

The title has room for improvements.  Both “procedures” and “e-government” is not well justified.  Maybe: ”ICT in rural areas from the perspective of dairy farming”

  1. Introduction

“The Digital Agenda for Europe”, should be properly referenced.

Connectivity may bring new opportunities to rural areas. New radio-based technologies may create the necessary connectivity at reasonable costs.  Internet access may even be more important for rural settlements, where online services may reduce the need for travel to administrative centers.

  1. Methodology

It is unclear how 56 search queries were made from 8 ICT terms and 26 rural terms. Also, it is unclear how the terms were chosen. The 8 ICT terms are connected to “inclusion and connectivity”, but it is not clear how the remaining terms are connected to subsections 3.2-3.9. Why “electronic government relationship” and not “electronic government”. What about “e-government”?  What were the final restrictions adopted for selection?

  1. Results

How was the 9 main topics select?

Table 1 has a number of errors.  [14] is duplicated for “smart farming”, [101] is duplicated for “market”, [117] is duplicated for “management and counselling”, and [29] is duplicated for “administration”.  References [6],[7],[8], and [125] have not been assigned to any thematic area.  There are some overlaps, but the overlap between “connectivity & inclusion” and “administration” consists of six papers, this should be explained.  Maybe it is a problem with the classification. All other overlaps are one or two papers.

Table 1 has some spelling errors: Conectivity -> Connectivity, Counceling -> Councelling.

Inconsistent use of capital letters: “Management & Counceling” vs. “Innovation and technology”.

Figure 1 is wrong due to wrong due to errors in Table 1. (Duplicates). How should multiple papers be calculated? If some papers are counted twice, they influence the percentages.  The best would probably be to split the value for papers appearing in two thematic areas. The aggregated percentages should be 100%, not 102%.

The following subsections do not always focus on dairy farming and livestock management. Many references point to general problems.  Some specific comments:

3.1 Connectivity and Digital Inclusion

Anyway, why do people with good connection not log in?  This question was raised, but not answered.

3.4 Smart Farming

This subsection could mention some of the advances actually used in livestock management such as:

Use of GPS/radio locators for livestock location monitoring

Use of virtual fences (a virtual fence provides a small electric shock when the animal reaches the perimeter of the grassland).

3.8 Management and counselling

Inconsistent use of capital letter in subheading. Should be “Management and Councelling”

  1. Discussion

Research should be used as a mass noun.  Researches should not be used.

References

Reference [125] seems to be never used.

Author Response

Dear Sir or Madam,

We would like to thank you for your time and kind contributions concerning this paper. In the next lines we will try to give an answer to your notes and comments:

In methodology: “Wit is unclear how 56 search queries were made from 8 ICT terms and 26 rural terms. Also, it is unclear how the terms were chosen. The 8 ICT terms are connected to “inclusion and connectivity”, but it is not clear how the remaining terms are connected to subsections 3.2-3.9. Why “electronic government relationship” and not “electronic government”. What about “e-government”?  What were the final restrictions adopted for selection?”. We understand your observations and as a way of clarifying these questions we have added a conceptual map in relation to the farmers’ needs and expectations in the section of results, since these are the bases upon which we have developed our searches. In terms of the e-government, we have verified that it was a typographical mistake which has already been modified.

In results: “How was the 9 main topics select?”. As an answer to this question we have clarified the section of results and as we have mentioned above, we have included a conceptual map as a base for our searches and divisions by areas.

“Table 1 has a number of errors.  [14] is duplicated for “smart farming”, [101] is duplicated for “market”, [117] is duplicated for “management and counselling”, and [29] is duplicated for “administration”.  References [6],[7],[8], and [125] have not been assigned to any thematic area.  There are some overlaps, but the overlap between “connectivity & inclusion” and “administration” consists of six papers, this should be explained.  Maybe it is a problem with the classification. All other overlaps are one or two papers”. Concerning this point, we did not consider doing a univocal distribution by areas, since a close reading of the papers provided answers for more than one section.

“Table 1 has some spelling errors: Conectivity -> Connectivity, Counceling -> Councelling”. As an answer to your observations, we have revised and modified these typographical mistakes both in table 1 and in the Graphical Abstract.

“Inconsistent use of capital letters: “Management & Counceling” vs. “Innovation and technology””. We have carried out a detailed inspection of the paper in order to address these issues.

“Figure 1 is wrong due to wrong due to errors in Table 1. (Duplicates). How should multiple papers be calculated? If some papers are counted twice, they influence the percentages.  The best would probably be to split the value for papers appearing in two thematic areas. The aggregated percentages should be 100%, not 102%.” The percentages have been calculated in relation to the total number of contributions by areas (the total number of papers analysed has not been used for this calculation). Concerning the summation of the percentages, it has been revised and modified according to your notes.

In Connectivity and Digital Inclusion: “Anyway, why do people with good connection not log in?  This question was raised, but not answered”. As an answer to your question, it is a knowledge gap for future research, as it has been mentioned in the conclusions. This paper is the first step of my PhD, which is oriented towards learning the current state of the relation between dairy farms and administrative authorities. We are currently working on two papers which are meant to cover the paths and limitations suggested in the conclusions of this very paper. We can say that after having conducted several surveys to dairy farmers in the region of Galicia (in the northwest of the Iberian Peninsula), we have confirmed that in rural areas the internet connection is still very poor. Besides, it is surprising to find the high number of users which, despite having an internet connection, showed reluctance when it comes to communicate with the public administration via e-governemt. We have detected many shortcomoings on the web platforms offered to the users by the administration, and new knowledge gaps to address applications which meet the users’ expectations.

In Smart Farming: “This subsection could mention some of the advances actually used in livestock management such as: Use of GPS...Use of virtual fences...”. We considered this a very wise and helpful comment, therefore we have included new contributions in relation to these terms in section IoT, as can be seen in citations 102-108.

In discussion: “Research should be used as a mass noun.  Researches should not be used”. We completely agree with this point and we have made this change

In references: “Reference [125] seems to be never used”. This is a typographical mistake which we have already solved, this citation now being [141].

Reviewer 2 Report

The article has structural problems.

Authors use very short paragraphs, 2 or 3 lines. Please add paragraphs so that the same idea is not subdivided and is presented consistently and without interruption. The paragraph should only change when the argument changes.

In the Introduction, a more detailed analysis of the study's contribution to the state-of-the-art of subject is lacking. For example, the dairy industry clearly has two configurations, SFSC (short chains) and large organized distribution. ICT contributions are different in each one. It is necessary to define SFSC (use https://doi.org/10.1016/j.jclepro.2017.09.235 as a reference) and large organized distribution (use https://doi.org/10.1016/j.jclepro.2017.08.188 and https://doi.org/10.1016/j.jclepro.2020.120254 as references). Then it addresses what each should expect from ICT.

A chapter on literature review is missing. The main concepts that the article uses must be explained in this chapter.

Chapter 3 is too poor for a scientific article. The SLR methodology should be explained in detail here. What was the criterion used to separate articles into groups? A criterion that can be useful is the analysis of clusters. Observe and do the same, or at least use an equivalent method. Just reporting the result is not enough. An implications section is missing (who gets what and why with your study). This section can be incorporated into the last section of your article. In this case, this section should be called Final Remarks. Please add the next research steps. What new research trails has your study opened up?

The number of references is low. Some references on open innovation are necessary as your study strongly relies on it. You can look in the journal J. Open Innov. Technol. Mark. Complex., especially articles that involve production chains like https://doi.org/10.3390/joitmc6040179.

Author Response

Dear Sir or Madam,

We would like to thank you for your time and kind contributions concerning this paper. In the next lines we will try to give an answer to your notes and comments:

Authors use very short paragraphs, 2 or 3 lines. Please add paragraphs so that the same idea is not subdivided and is presented consistently and without interruption”. We understand your comment and therefore we have restructured the information in the document using larger paragraphs.

In the Introduction, a more detailed analysis of the study's contribution to the state-of-the-art of subject is lacking. For example, the dairy industry clearly has two configurations, SFSC (short chains) and large organized distribution. ICT contributions are different in each one. It is necessary to define SFSC…and large organized distribution... Then it addresses what each should expect from ICT”. We consider this a very wise and interesting contribution and we would like to thank you for the papers suggested. We have added new information concerning these notes, as can be seen in the citations 19-25.

A chapter on literature review is missing. The main concepts that the article uses must be explained in this chapter”. We understand your observation but after revising the literature, we found several alternatives concerning how to address this issue, yours being one of them. Another alternative, which is the one chosen for this paper, was to define the concepts at the beginning of each section. We considered that this was an appropriate option, and changing it would mean a change in the structure of the whole document.

“Chapter 3 is too poor for a scientific article. The SLR methodology should be explained in detail here. What was the criterion used to separate articles into groups? A criterion that can be useful is the analysis of clusters. Observe and do the same, or at least use an equivalent method. Just reporting the result is not enough”. We understand and agree with your comments, therefore we have developed a conceptual map concerning the dairy farmers’ needs and expectations, since this is the basis of our search. We have also clarified the results with the aim of addressing the points you mention.

“An implications section is missing (who gets what and why with your study). This section can be incorporated into the last section of your article. In this case, this section should be called Final Remarks. Please add the next research steps. What new research trails has your study opened up?”. This is a very interesting contribution and we have added a section of conclusions in order to address these points, with new research paths.

“The number of references is low”. We understand your observations, but the scientific field which addresses this paper is not an area in which there are many scientific contributions. As stated in the methodology, we revised 1683 papers, of which we selected approximately 10%. We have added new contributions, obtaining a total of 169 papers revised (initially we had 147).

Reviewer 3 Report

Dear Colleagues,

I think that you need to take our section on highlights. You need to add the research question and aim of your research. In the abstract, you need to change this document to this article or paper. In your methodology is unclear when you made your search and If I try to find out if I find the same number of articles. Unclear which gap you try to close by doing your research. Also, I lacked a very important trajectory in the development of the internet and e-government. It is m-government. You can read more Societies | Free Full-Text | Upstream Social Marketing for Implementing Mobile Government (mdpi.com) 

In the section of the discussion, it will be good to have the table where you compare your findings according to some criteria. Also, you need a section of conclusions. A section on limitations is needed.

Best regards

Author Response

Dear Sir or Madam,

We would like to thank you for your time and kind contributions concerning this paper. In the next lines we will try to give an answer to your notes and comments:

 “You need to add the research question and aim of your research”. Taking into account your comment, we have modified the abstract, stating the aim of our research and clarifying the different contributions by areas. In the same way, we have also clarified the objective in the section of introduction.

In the abstract, you need to change this document to this article or paper”. We agree and we have made this change.

In your methodology is unclear when you made your search and If I try to find out if I find the same number of articles. Unclear which gap you try to close by doing your research”. Our objective is to delve into the relationship between dairy farmers and the administrative authorities via e-government, comparing this area to other 8 concerning the farmers’ needs and expectations in relation to the ICT in different fields of their business. We understand your observations, but the scientific field which addresses this paper is not an area in which there are many scientific contributions. As stated in the methodology, we revised 1683 papers and we cited a total of 169 (mostly from the last three years), which mean approximately 10% of the total.

“Also, I lacked a very important trajectory in the development of the internet and e-government. It is m-government”. We considered this a very interesting contribution and as such, we have addressed it in section 3.9 about Public Administration. The new papers correspond to citations 161-166.

“In the section of the discussion, it will be good to have the table where you compare your findings according to some criteria”. This is a very sensible change, therefore we have placed the table in the section of discussion, as you suggest.

“You need a section of conclusions. A section on limitations is needed”. We understand your comment and we have addressed this issue adding, as you recommend, a section of conclusions at the end of the document in which we also try to deal with the limitations or gaps found in this paper, leading to future research.

Round 2

Reviewer 2 Report

Authors hava satisfactorily addressed most issues

Reviewer 3 Report

Thank you!